# Metavalently bonded tellurides: the essence of improved thermoelectric performance in elemental Te

Decheng An [1], Senhao Zhang[2], Xin Zhai[3], Wutao Yang[1], Riga Wu[2], Huaide Zhang[2], Wenhao Fan[4], Wenxian Wang[4], Shaoping Chen[4], Oana Cojocaru-Mirédin[5], Xian-Ming Zhang [1,4] ✉, Matthias Wuttig [2,6] ✉ & Yuan Yu [2] ✉

Elemental Te is important for semiconductor applications including thermoelectric energy conversion. Introducing dopants such as As, Sb, and Bi has been proven critical for improving its thermoelectric performance. However, the remarkably low solubility of these elements in Te raises questions about the mechanism with which these dopants can improve the thermoelectric properties. Indeed, these dopants overwhelmingly form precipitates rather than dissolve in the Te lattice. To distinguish the role of doping and precipitation on the properties, we have developed a correlative method to locally determine the structure-property relationship for an individual matrix or precipitate. We reveal that the conspicuous enhancement of electrical conductivity and power factor of bulk Te stems from the dopant-induced metavalently bonded telluride precipitates. These precipitates form electrically beneficial interfaces with the Te matrix. A quantum-mechanical-derived map uncovers more candidates for advancing Te thermoelectrics. This unconventional doping scenario adds another recipe to the design options for thermoelectrics and opens interesting pathways for microstructure design.

Since the discovery of bulk tellurium (Te) as an elemental semiconductor in the early 1930s, doping−the incorporation of impurities into crystalline solids−has been contemplated in tailoring its electrical conductivity[1,2]. The dopant-induced electronic property manipulation is widely believed as a prerequisite to advance thermoelectric performance. This quantity is determined by a dimensionless figure-of-merit, $zT = S^2\sigma T/\kappa$, where $S$, $\sigma$, $T$, $\kappa$ denote the Seebeck coefficient, the electrical conductivity, the absolute temperature, and the thermal conductivity, respectively. Over the past decade, elemental Te with intrinsically low thermal conductivity has emerged as a high-

performance thermoelectric energy conversion material within the 450–600 K range upon introducing proper dopants[3–6]. As summarized in Fig. 1a, the functional effects of various dopants in Te-based thermoelectrics can be classified into two categories, in which the compositions corresponding to a significant enhancement of $zT$ seem to invariably contain group-VA-elements (i.e., As, Sb, and Bi; green region). By contrast, introducing other dopants such as Se, Ag, and Zn (red region) leads to marginal improvements in the thermoelectric properties. From the perspective of carrier transport, there was also a prominent difference (around two orders of magnitude) in carrier

[1]College of Chemistry, Taiyuan University of Technology, 030024 Taiyuan, China. [2]Institute of Physics (IA), RWTH Aachen University, Sommerfeldstraße 14, 52074 Aachen, Germany. [3]School of Electronic Science & Engineering, Southeast University, 210096 Nanjing, China. [4]Key Laboratory of Interface Science and Engineering in Advanced Materials, College of Materials Science and Engineering, Instrumental Analysis Center, Taiyuan University of Technology, 030024 Taiyuan, China. [5]Department of Sustainable Systems Engineering (INATECH), Albert-Ludwigs-Universität Freiburg, 79110 Freiburg, Germany. [6]Peter Grünberg Institute (PGI 10), Forschungszentrum Jülich, 52428 Jülich, Germany. ✉e-mail: zhangxianming@tyut.edu.cn; wuttig@physik.rwth-aachen.de; yu@physik.rwth-aachen.de

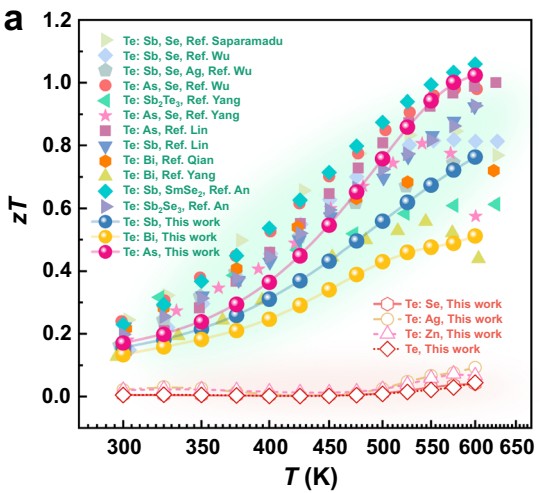

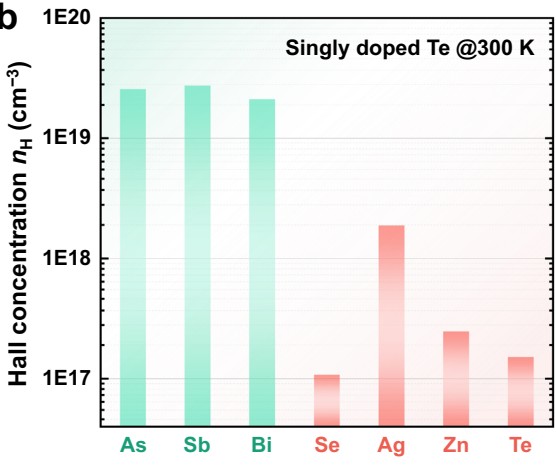

**Fig. 1 | Thermoelectric properties of p-type Te doped by various dopants.**
**a** Temperature-dependent $zT$ for polycrystalline Te doped with (green region) and without (red region) group VA elements (i.e., pnictogens: As, Sb, and Bi) (see Supplementary Fig. S1–S4 for more detailed transport properties). To enable uni-variate analysis, the content of dopants was uniformly set at 2% in this work, regardless of their difference in optimum doping level. Only compositions with experimental values in the literature are collected here[3,8,10–12,56,57,59,67,76–78]. **b** Room-temperature Hall carrier concentration ($n_H$) for single-element-doped bulk Te systems containing pnictogens (green), compared with that of pristine Te and other non-pnictogen samples (red). Note that the optimal $n_H$ level is about $1 \times 10^{19}$ cm$^{-3}$ in Te thermoelectrics[7].

concentration for bulk Te doped with (green) and without (red) group-VA-elements (Fig. 1b). This poses an intriguing question: why are group-VA-elements (As, Sb, and Bi) the high-efficiency p-type dopants for elemental Te, while others are not?

In some pioneering density functional theory (DFT) studies investigating hole-doped bulk Te, an improved hole concentration has been predominantly attributed to the complex band structure upon Sb or Bi doping[7,8]. However, the accuracy of such a calculation is questionable owing to the lack of rigorous estimates on real doping concentration in the Te system. Aside from S and Se lying in the same chalcogen group as Te, almost all other dopants possess extremely low solubility (<0.1 at.%) in Te according to the Te-based binary alloy phase diagrams[9]. This point, in turn, is difficult to reconcile with the experimental observation that thermoelectric parameters can be meticulously controlled through the manipulation of initial doping content (ranging from 0.1 to 10 at.%) in the case of As-[10], Sb-[11], or Bi-doped[12] Te. To date, there are no detailed studies that can account for the complexity of structure–performance relationships since the microstructures are often investigated on a local scale while the transport properties are measured on a much larger scale. It is thus necessary to measure the transport properties on a micrometer scale as well to enable an in-depth understanding of the link between dopant-induced microstructures and the resulting thermoelectric properties.

Herein, we develop a structure–property correlation method that enables precise measurement of the impact of individual microstructures on the local electrical transport properties. We discover that dopant-induced telluride precipitates dominate the transport properties, whereas the small dopant content in the matrix provides a marginal contribution to electronic conductivity. This conclusion revises our understanding of the impurity doping effect, which is usually assumed and/or shown to primarily tailor the properties of the matrix[13–19]. By combining bond-rupture behavior characterizations using atom probe tomography (APT) and DFT calculations, the internal "metavalent bonding (MVB)" mechanism of these precipitates is further clarified. The distinct property portfolio of MVB compounds and their electrically beneficial interface with the Te matrix will be shown to be crucial to empowering high-performance Te thermoelectrics. Moreover, we provide a material design map related to the fundamental chemical bonding information for forecasting potential dopant species. Upon experimental verifications, we find that valid new telluride dopants are all characteristic of metavalent solids.

## Results

### Structural and compositional characterization

We first examined the phase compositions of the as-prepared $Te_{0.98}M_{0.02}$ (M = As, Sb, and Bi) using X-ray diffraction (XRD) measurements (Fig. 2a–c). In addition to the conspicuous Te matrix phase with a trigonal structure (space group $P3_121$), we observed the secondary-phase Bragg reflections belonging, respectively, to the rhombohedral $\beta$-$As_2Te_3$ (space group $R-3m$), $Sb_2Te_3$ (space group $R-3m$) and $Bi_2Te_3$ (space group $R-3m$). This observation agrees well with prior studies on the singly doped Te samples[11]. Moreover, we found that the peak positions of the Te phase did not shift upon increasing the nominal dopant contents from 0 to 15 at.% (Supplementary Fig. S5a–c), indicating an absence of appreciable changes in the lattice parameters upon As, Sb, or Bi doping. Employing scanning electron microscopy (SEM) coupled with energy-dispersive X-ray spectroscopy (EDS), we further characterized the compositional distribution of this dual-phase structure, as shown in Fig. 2d–f. Despite the trace doping amount, all three doped samples exhibit large amounts of micrometer-scale telluride precipitates dispersed in the Te matrix, consistent with our XRD analyses. Unfortunately, doping elements were not detected in the matrix regions due to the limited spatial and chemical resolution of EDS. Therefore, we have turned to APT to provide a quantitative compositional analysis at a near-atomic scale[20–23] (Fig. 2g–i). APT investigations revealed extremely low solute concentrations (i.e., solubilities) in the matrix (on average, 0.06 at.% As, 0.03 at.% Sb, 0.09 at.% Bi). This is in line with the equilibrium phase diagrams of the M (As, Sb, Bi)−Te system, which show that the thermodynamic boundary between these telluride phases and the Te phase, namely the solubility line, approaches pure Te below 450 °C (see Supplementary Fig. S5d–f). Given that these dopants overwhelmingly form telluride precipitates instead of solute atoms in the Te matrix and the fraction of precipitates increases with increasing the doping content[11], it is thus natural to ask what role precipitates play in the improvement of thermoelectric performance of elemental Te.

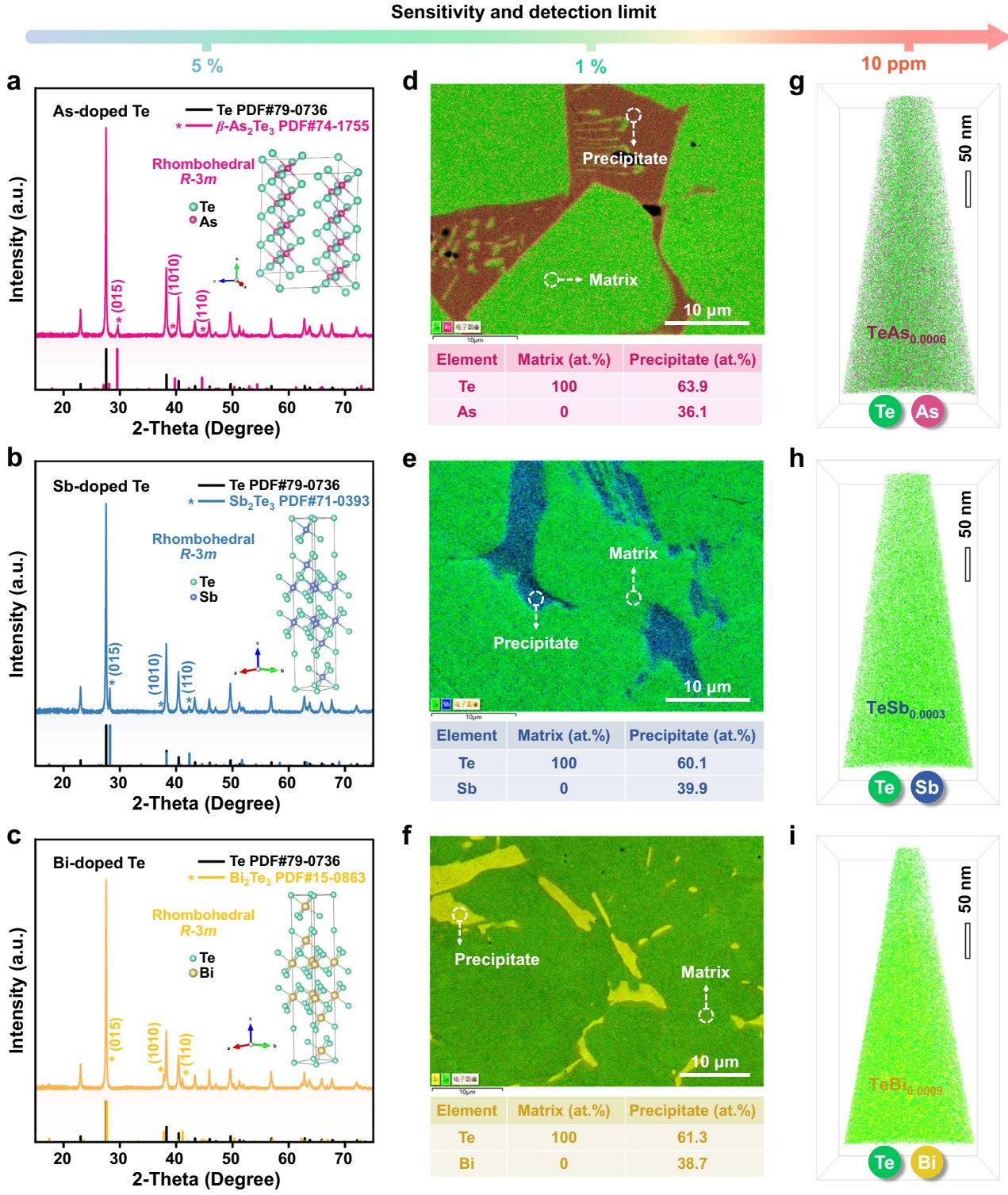

**Fig. 2 | Multiscale microstructure characterization.** Analytical techniques spanning multiple chemical sensitivities were used to investigate the microstructural features of three representative $Te_{0.98}M_{0.02}$ materials (M = As, Sb, and Bi). **a–c** The XRD patterns of the doped Te powders. **d–f** The SEM-EDS results of the polished doped samples, which show the presence of numerous embedded telluride precipitates (consisting of approximately 60 at.% Te and 40 at.% M, i.e., $M_2Te_3$-type, M = As, Sb, and Bi). **g–i** Distribution of elements in the Te matrix region of the doped samples characterized by APT with a much higher spatial resolution and chemical sensitivity (ppm level) than XRD and EDS. In the APT images, green, fuchsine, blue, and yellow dots represent Te, As, Sb, and Bi atoms, respectively.

## Local electrical conductivity measurement

To answer the above question, we first compared the bulk transport properties upon tuning the precipitate content. As seen in Fig. S6, over the temperature range of 300–600 K, all three series of samples yielded a visible variation in the electrical properties upon increasing the dopant concentration (equivalently the fraction of precipitates) after exceeding their solubilities, which alludes to the unique impact of precipitates on carrier transport. Notably, these tendencies are based

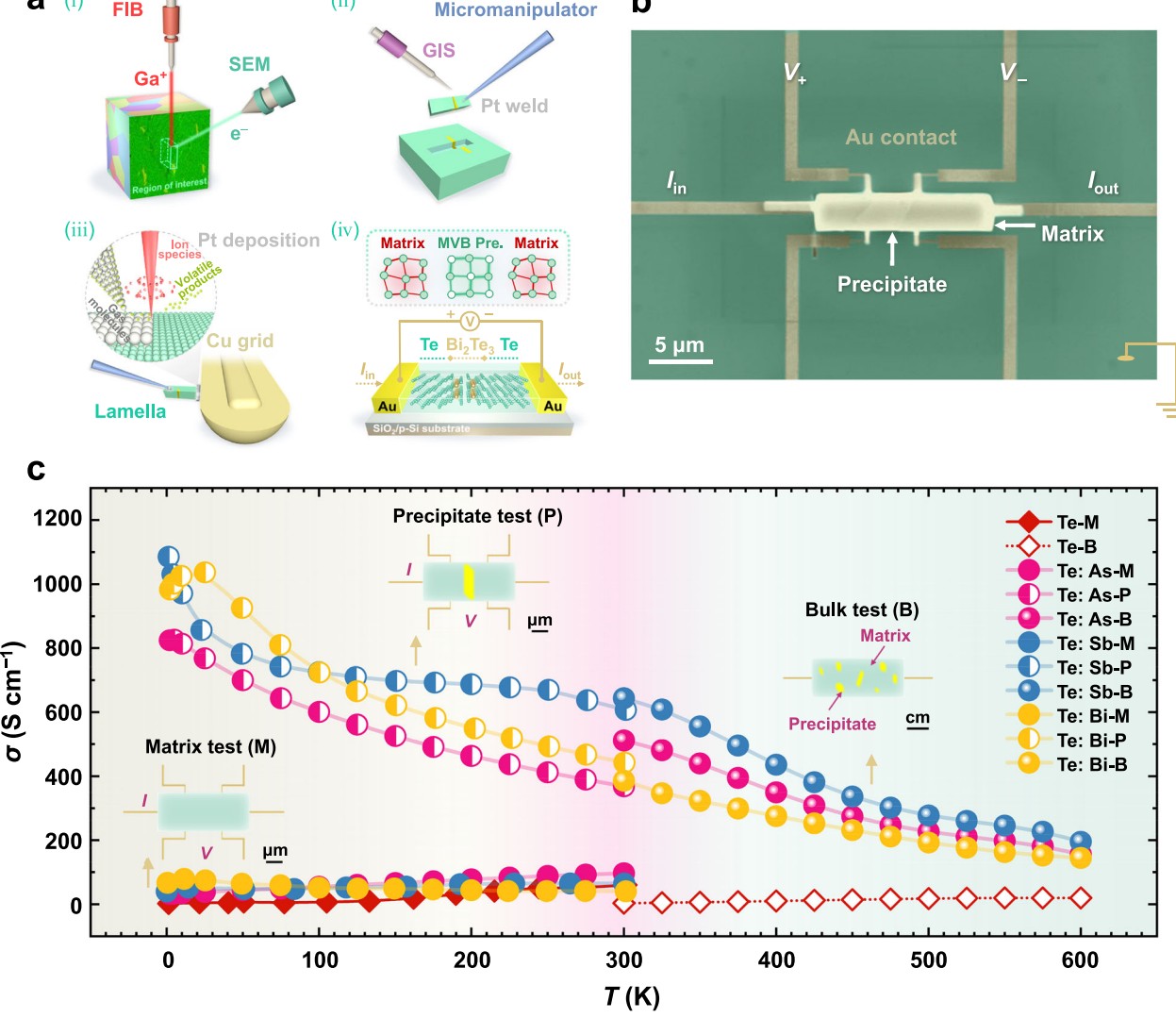

**Fig. 3 | Mesoscopic transport property measurement scheme. a** Schematic drawing of the fabrication process of a representative individual $Bi_2Te_3$ precipitate device: (i) trench milling around the target region of interest with the aid of FIB and SEM; (ii) the micrometer-sized lamella attached to the micromanipulator is lifted out using the in-situ gas injection system (GIS); (iii) the thinning process of the lamella that was secured to the side of the Cu grid post; (iv) chip assembly of the processed lamella containing the $Te/Bi_2Te_3$ heterostructure on a $SiO_2/Si$ substrate. **b** False-colored SEM image of the fabricated Hall bar device used to measure the conductivity $\sigma$ of the $Te–Bi_2Te_3$ heterophase region, in which the lamellar specimen is contacted to six Au electrodes (deep yellow). **c** Comparison of temperature-dependent local electrical conductivity measured on the As-, Sb-, and Bi-doped Te lamella samples (nominal composition $Te_{0.98}M_{0.02}$, M = As, Sb, and Bi) with and without telluride precipitates. Corresponding bulk electrical conductivities are measured for reference. Note that "M", "P", and "B" in the legend of Fig. 3c represent "matrix", "precipitate", and "bulk", respectively.

on routine macroscopic observations—that is, on measuring centimeter-scale specimens embodying average performance. Such experiments hence cannot provide insight on the role of precipitates and the doped matrix. We have addressed this challenge by independently measuring the electrical conductivities ($\sigma$) of polycrystalline-doped-Te samples with and without individual precipitates at the local micrometer length scale. As schematically sketched in Fig. 3a, site-specific microstructures (for instance, precipitates) can be identified under an SEM. After that, the precipitate will be included in the micrometer-sized lamella sample with defined dimensions prepared via a focused ion beam (FIB) lift-out methodology[24]. Figure 3b displays the as-assembled mesoscopic transport device with a Hall-bar configuration on the p-Si substrate with a $SiO_2$ capping layer. The position of an individual precipitate within the lamella lies in the middle of two electrodes. This geometry facilitates the measurement of the longitudinal electrical conductivity as a function of temperature

(10−300 K) utilizing a physical property measurement system (PPMS). Details are described in the "Methods" section and our previous work[24,25]. In this way, we also fabricate measurement devices incorporating only the Te matrix, i.e., without precipitates (see Fig. S7), as a control to extract the contribution of the precipitate to the transport properties.

Figure 3c plots the temperature-dependent electrical conductivity measured by the micro-scale Hall-bar device for samples with or without an individual precipitate. The conductivity data for bulk samples above room temperature are also presented for comparison. The pure Te bulk sample exhibits an intrinsic semiconducting characteristic as $\sigma$ decreases monotonically with reducing temperature ($T$). Interestingly, the micro-scale pure Te and all the doped samples without precipitates show a similar semiconducting behavior. We further verified that there was little change of electronic properties in the pnictogen-doped Te matrix (e.g., $\sigma \sim 43.7$ S cm⁻¹ at 200 K in the Bi-

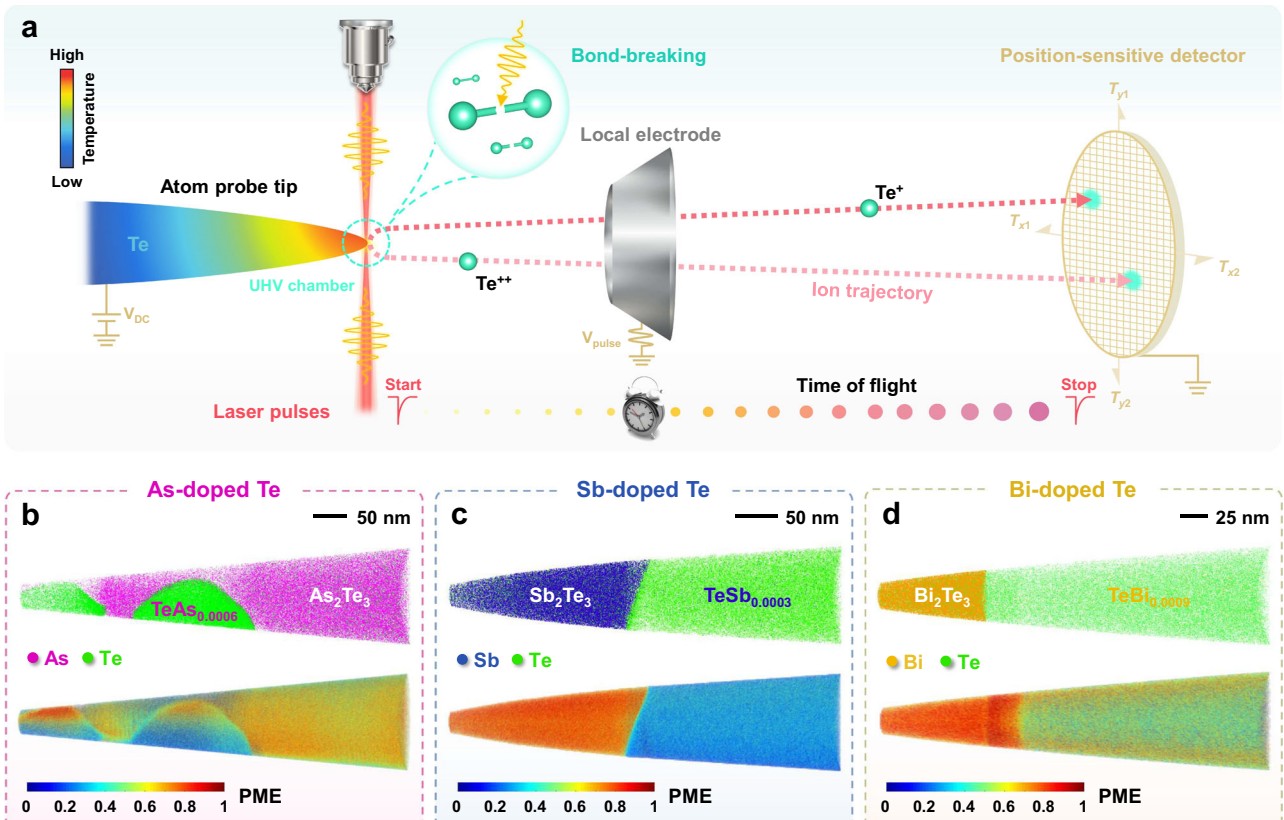

**Fig. 4 | Atom probe tomography technique used to investigate the bond-breaking behavior in doped polycrystalline Te. a** Schematic illustration of the laser-assisted atom probe tomography under ultra-high vacuum (UHV) conditions. Atoms at the tip-like shape specimen surface are field evaporated as ions by breaking the chemical bond in a high electric field ($10^{10}$ V/m) and then projected onto a position-sensitive detector[79]. These detected ions could be chemically identified on account of time-of-flight mass spectrometry. The reconstructed three-dimensional (3D) maps concerning Te (green), As (fuchsin), Sb (blue), and Bi (yellow) atoms obtained from **b** As-, **c** Sb-, and **d** Bi-doped Te APT specimens and the corresponding spatial distribution of the multiple events. Ions in the precipitate regions of rhombohedral $\beta$-$As_2Te_3$, $Sb_2Te_3$, and $Bi_2Te_3$ were preferentially evaporated with multiple ions, leading to >70% probability of multiple events (PME). The PME value represents the ratio of multiple events to total events. On the contrary, a single atomic or molecular ion in the matrix regions of doped Te was favorably field-evaporated per successful laser pulse with a PME value < 30%. A high PME value is one of the prevailing characteristics of materials employing MVB.

doped case) through the same lamella test, relative to those in the intrinsic matrix ($\sigma \sim 41.1$ S cm$^{-1}$ at 200 K). This is a direct manifestation of the ineffectiveness of the elemental doping effect of the Te matrix. In striking contrast, all three lamellae containing an individual telluride precipitate (namely $As_2Te_3$, $Sb_2Te_3$, and $Bi_2Te_3$) embedded in the Te matrix present an overall metallic-like nature with $\sigma$ decreasing upon increasing $T$. Their $\sigma$ values are almost two orders of magnitude higher than the corresponding values of the matrix. This finding indicates that the precipitates dominate the transport properties. The behavior observed is fundamentally different from that observed in more conventional thermoelectric semiconductors, where only heavily doped materials reveal degenerate conductivitiy[26]. Indeed, the discontinuity in $\sigma$ versus $T$ curves between the bulk and lamella samples at 300 K is also attributed to their differences in both the volume ratio of precipitate versus matrix and the related boundary defect density. This again underscores the prominent role of group V telluride precipitates in regulating Te-based thermoelectric materials.

## Chemical bonding mechanisms
To further understand the role of the precipitates, we have investigated them using APT to unveil their atomic arrangement and chemical bonding. Figure 4a schematically illustrates the basic structure and working principle of APT, and more details can be found in other literature[20,21,27]. Besides the outstanding capabilities in characterizing structural defects at the sub-nanometer scale in 3D, APT can also distinguish different chemical bonding mechanisms through bond-

breaking behavior[20,28–31]. It has been found that the probability of dislodging several fragments from the specimen under one successful laser pulse, i.e., the "probability of multiple events (PME)", is much larger in metavalently bonded compounds than that in other materials[28,29,32–35]. As shown in Fig. 4b–d, the matrix regions of As-, Sb-, and Bi-doped Te samples reveal an ordinary bond-breaking character with a moderate PME value (about 30–40%). This is also coincident with the evaporation behavior for undoped Te (see Supplementary Fig. S8a). In stark contrast, for the precipitate regions identified as $As_2Te_3$, $Sb_2Te_3$, and $Bi_2Te_3$, much higher PME values (around 70–80%) are observed. This unusually high PME value so far has only been found in materials utilizing metavalent bonding (MVB)[29,34]. This is also in line with our previous work that these sesqui-tellurides employ MVB[36]. Metavalent bonds (MVB) are located in a map spanned by two quantum-chemical bonding descriptors between covalent and metallic bonds[37]. They are best described by a situation where adjacent atoms share half an electron pair (one electron) to form a σ-bond, i.e., a "two center–one electron" (2c–1e) bond[38]. This scenario is fundamentally different from ordinary covalent bonds, where adjacent atoms are held together by an electron pair, i.e., a 2c–2e bond. Metavalent bonds also differ from metallic bonds, where the electrons are rather delocalized. Yet, MVB is not a combination of metallic and covalent bonds. It is instead derived from the competition between electron localization, as in covalent and ionic bonds, and electron delocalization as in metallic bonds[39]. This is demonstrated by the distinctively different properties of materials utilizing these different bonding mechanisms[39].

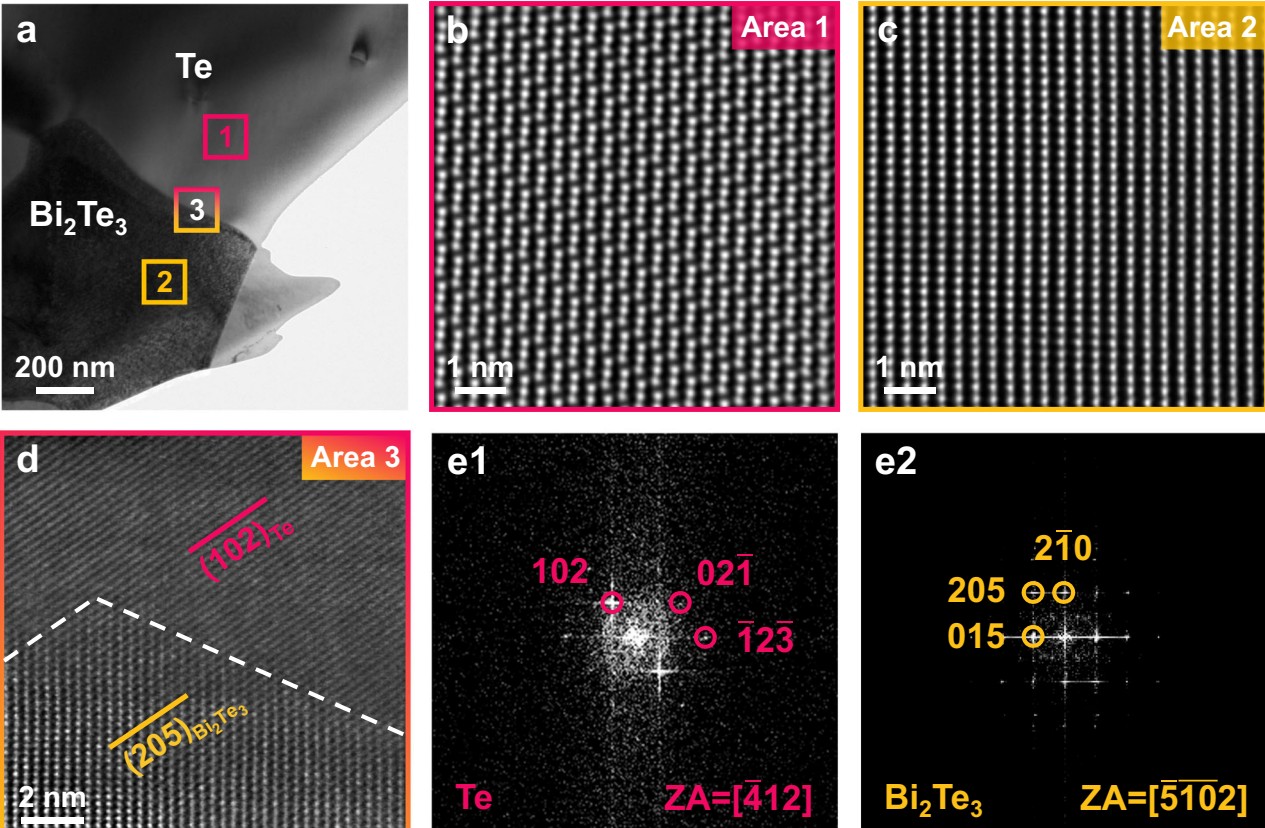

**Fig. 5 | Interfacial characterizations of the Bi2Te3/Te heterostructure.** **a** Representative TEM image of the $Bi_2Te_3$ precipitate in the Bi-doped Te matrix. Atomic-resolved HAADF-STEM images for **b** Te, **c** $Bi_2Te_3$, and **d** the interface domain, respectively. In **e1** and **e2**, fast Fourier transform (FFT) patterns are calculated from the region indicated by the red and yellow marks in **d**, respectively.

Metavalent solids, including PbTe, show a unique portfolio of properties, such as a large optical dielectric constant which describes the electronic polarizability, a high Born effective charge which describes the chemical bond polarizability, and a high mode-specific Grüneisen parameter which describes the lattice anharmonicity. This unique property portfolio is not formed by a combination of corresponding properties of covalent and metallic solids. In addition, metavalent solids show an abnormal bond rupture with a large value of PME, as illustrated above. In contrast, metals and covalently bonded compounds show PME values below 40%[28]. The high PME value of metavalent solids is a unique identifier of this bond type. More detailed discussions on the correlation between the high PME and the MVB mechanism can be found in other studies[20,24,28,29,32]. We also noticed that the interfaces between Te and precipitates show much higher PME values than the Te matrix. This could be an indication of the special interface properties.

The valence electrons in MVB are in a competitive state between localization and delocalization[39], which enables the MVB system characteristic of incipient metals with a large electrical conductivity. This can be directly seen in the micro-device measurement results in Fig. 3c. All the lamellae, including MVB precipitates, show a metallic transport behavior and conductivity values several orders of magnitude higher than the pristine Te and the matrix of doped Te. Additionally, the MVB compounds generally show a large valley degeneracy due to their high crystal symmetry and a small band effective mass due to the large dispersion of energy bands. These factors together lead to a high power factor (PF = $S^2\sigma$)[29,32,40]. More detailed discussions can be found in Fig. S9. Furthermore, metavalent solids are characterized by half-filled σ-bonds, i.e. a bond order of 0.5 only, compared with a bond order of 1 for ordinary covalent bonds. This leads to weaker and softer

bonds, consistent with larger bond lengths. Employing bond length–bond strength correlations supports the existence of soft bonds in MVB materials. As a result, a low phonon group velocity is obtained in MVB materials[32]. On the other hand, the delocalized system of $p$-electrons in MVB materials is unstable against transverse optical modes, inducing a softening and large anharmonicity of optical phonons[41]. This finding is closely related to the large Born effective charge. As a result, the phonon relaxation time is greatly reduced due to anharmonic phonon scattering. These two factors contribute to the low intrinsic thermal conductivity of MVB materials[32,41]. This manifests that these tellurides are intrinsically good thermoelectric materials, as has also been proved in their single-phase bulk materials[42–45]. It is very striking that these MVB tellurides as second phases can substantially increase the overall thermoelectric performance of Te-based composites.

## Interfacial structure and charge transfer

The superior thermoelectric properties of MVB tellurides are still not enough to explain the overall improved performance of the composite since the volume fraction of these precipitates is rather low, given the nominal composition of 2% dopants. Hence, it is crucial to determine the impact of metavalent precipitates on the overall material performance. Structural characterizations, as shown in Fig. 2, do not reveal a percolation network of these precipitates. Therefore, the interface between the matrix and the precipitate must play an important role in the transport properties of the entire sample. Figure 5a showcases a low-magnification transmission electron microscopy (TEM) image of a representative $Bi_2Te_3$/Te composite. We have further examined the atomic structure of Te, $Bi_2Te_3$, and their interface, employing high-angle annular dark-field scanning transmission electron microscopy

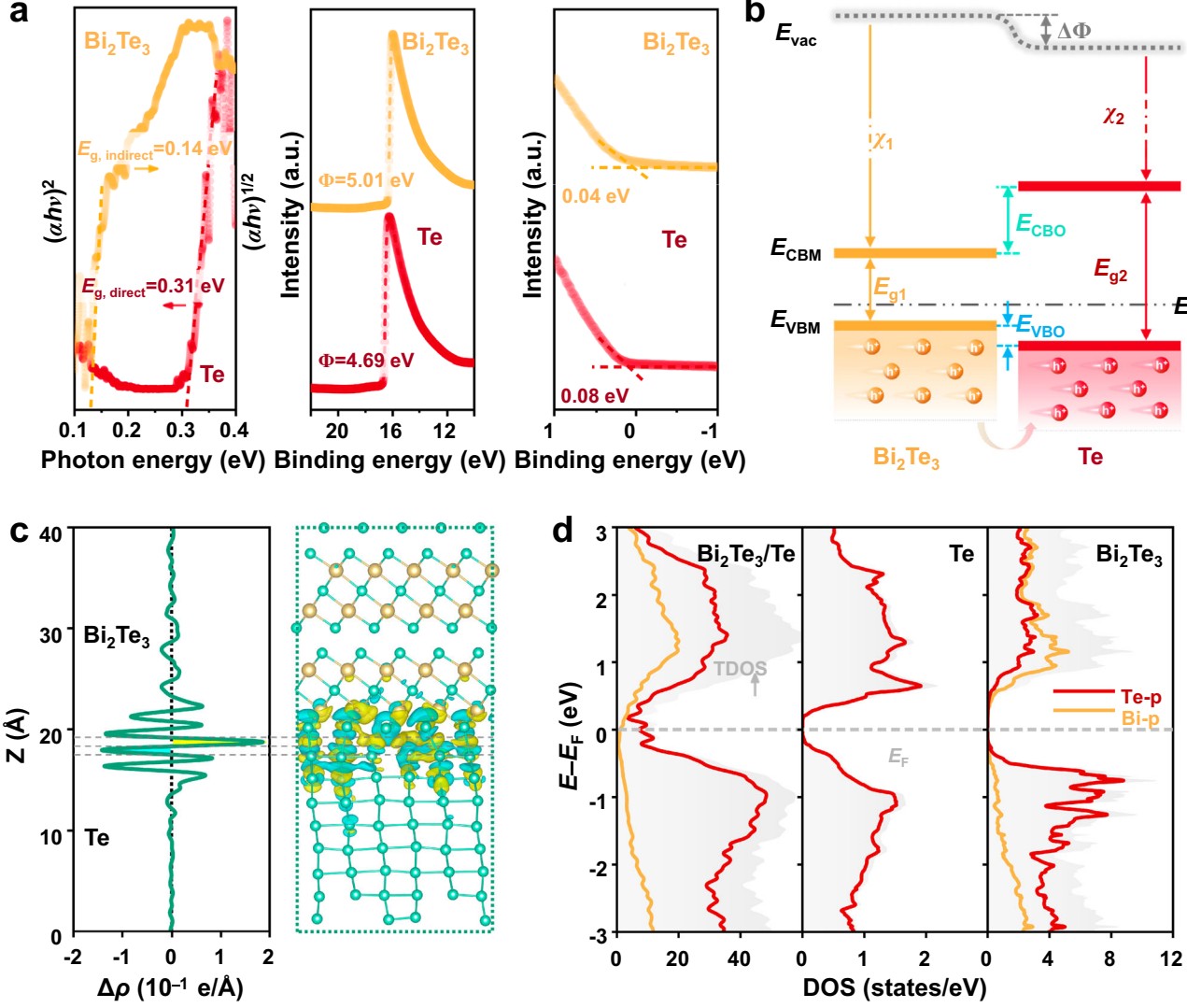

**Fig. 6 | Electrical characteristics of the Bi₂Te₃/Te interface. a** Optical bandgap, work functions, and valence band maximum (VBM) levels of Bi₂Te₃ and Te. The band gap values were determined from Tauc plots of optical absorption coefficient $(\alpha h\nu)^{1/n}$ versus photon energy $h\nu$ ($h$ is the Planck constant and $\nu$ is the frequency of the photon), using Fourier transform infrared spectrometer (FTIR) measurements, where the value of $n$ is taken to be 1/2 and 2 for indirect-gap[80–82] Bi₂Te₃ and direct-gap[51,83] Te, respectively. The work functions and VBM positions for Bi₂Te₃ and Te are extracted from ultraviolet photoelectron spectroscopy (UPS) spectra. **b** Schematic showing the experimentally determined energy-level alignment at the

Bi₂Te₃/Te hetero-interface. $E_{vac}$ vacuum level, $E_{CBM}$ conduction band minimum, $E_{VBM}$ valence band maximum, $\Delta\Phi$ work function difference between Bi₂Te₃ and Te, $\chi$ electronic affinity, which can be calculated by the following formula: $\chi = E_{vac} - E_{CBM}$; $E_{CBO}$ conduction band offset, $E_{VBO}$ valence band offset. **c** 3D charge density difference of the Bi₂Te₃/Te interface and corresponding planar-averaged differential charge density. The cyan and yellow areas denote electron depletion and accumulation, respectively. **d** Projected density of states (PDOS) of Bi₂Te₃/Te, bulk Te, and Bi₂Te₃. The Fermi level was set as zero.

(HAADF-STEM). Both Te (Fig. 5b) and Bi₂Te₃ (Fig. 5c) possess perfect lattices with no visible defects. This conclusion is supported by geometrical phase analysis (GPA) maps (see Fig. S10 for details), where no lattice strains are observed. Meanwhile, we observe a very smooth and continuous extension of atomic arrangements between the (102) plane of Te and the (205) plane of Bi₂Te₃, as demonstrated in Fig. 5d, e1, and e2. This indicates a high degree of lattice matching between these two phases, as further validated by fast Fourier transform (FFT) and corresponding inverse FFT patterns (Supplementary Fig. S11). Such a good registry requires strong interactions across the interface. The strong coupling between the dopant phase and the Te host requires metavalent bonds in the sesqui-chalcogenide precipitate. This is conducive to good charge transport and thus crucial to improving the properties of the composite material[46–49].

To shed further light on the underlying mechanisms of interface contribution to the thermoelectric performance, we then focus on the

charge transfer behavior across the interfaces. We have measured the optical band gap ($E_g$), the work function ($\Phi$), and the valence band maximum energy ($E_{VBM}$) in the case of individual Te and Bi₂Te₃ (Figs. 6a and S12). To be specific, we detected a direct $E_g$ of 0.31 eV and a $\Phi$ of 4.69 eV for Te, and we found the values of 0.14 and 5.01 eV for indirect $E_g$ and $\Phi$ of Bi₂Te₃, respectively. These values are consistent with results previously reported within the experimental uncertainty: 0.30[50] and 4.59 eV[51] for Te, as well as 0.13[52] and 4.83 eV[53] for Bi₂Te₃, respectively. In addition, the determined VBM of Te is 0.04 eV below that of Bi₂Te₃. These results allow us to build the electronic band structure diagrams for the Bi₂Te₃/Te interface, as illustrated in Fig. 6b. Evidently, a clear type-I (i.e., straddling gap) valence band alignment is obtained at the interface. This near-ideal band alignment will ensure a small barrier for hole transport (in terms of hole injection into Te). It is noted that this finding is based on the assumption that interfacial defects induced by lattice mismatch do not pin the Fermi energy. To

this end, we further used DFT calculations to validate the above arguments. A 2% lattice mismatch between Te (102) and $Bi_2Te_3$ (001) is built in the DFT-relaxed structures. As evident in Fig. S13, the calculated $\Phi$ of the $Bi_2Te_3$/Te interface (4.86 eV) is found between Te and $Bi_2Te_3$. Consequently, when both phases closely contact to form the interfacial structure, the electrons spontaneously migrate from Te to $Bi_2Te_3$ until the Fermi level reaches an equilibrium. This is supported by the plane-averaged charge density difference simulation in Fig. 6c. It is seen that the electrons are depleted at the Te side (cyan region) and accumulated at the $Bi_2Te_3$ side (yellow region), indicative of the p-type doping to Te. The $Bi_2Te_3$/Te interface possesses a higher density of state (DOS) near the Fermi level compared to Te and $Bi_2Te_3$ (Fig. 6d). Particularly, the states across the Fermi level for $Bi_2Te_3$/Te are highly suggestive of a metallic feature. This facilitates a significantly boosted conductivity of the interface, in line with the experiments presented in Fig. 3c. Metavalently bonded materials all possess small bandgaps and moderate electrical conductivities ($10^2$–$10^4$ S cm$^{-1}$) with a comparatively high carrier concentration[32]. As a result, p-type MVB telluride precipitates such as "sister materials" $\beta$-$As_2Te_3$, $Sb_2Te_3$, and $Bi_2Te_3$, can act as hole carrier reservoirs, offering additional holes to the undoped Te matrix while the whole system is applied under an exterior voltage or thermal gradient. MVB telluride/Te interfaces tend to form a type-I heterophase structure with a small valence band offset ($E_{VBO}$) owing to the slightly smaller bandgaps of MVB tellurides relative to Te (see Fig. S14 for details). For example, $Bi_2Te_3$/Te shows an $E_{VBO}$ of only 0.04 eV. In contrast, the $E_{VBO}$ value is 0.46 eV for the interface between ZnTe and Te (Fig. S15), which is too large to form valence band alignments. Owing to this favorable energy level alignment in conjunction with efficient interfacial charge transfer between MVB precipitates and matrix phases, the bulk carrier concentration and electrical conductivity can be tailored by introducing MVB group V tellurides in Te.

## Screening and verification of novel Te thermoelectrics based on a bonding map

We next provide a quantum-mechanical-based map for bonding in solids to further screen desired dopants (or more specifically, their MVB tellurides) for the Te system. The formation of different bonds between atoms can be attributed to the different behavior of valence electrons, such as electron transfer as in ionic bonds or electron pair formation as in covalent and metallic bonds[38]. Using quantum-mechanical calculation tools, the number of electrons transferred (ET) from one atom to its neighboring atom and the value of electrons shared (ES) between two neighboring atoms can be obtained[37]. These two values (ES and ET) are excellent descriptors of chemical bonds. Thus, a 2D map spanned by ES and ET can be established, where different types of chemical bonds are located in well-defined regions of the ES-ET map. MVB prevails in a region with a small ET and an ES value close to one. This is also in line with the nature of MVB as a "two center–one electron" (2c–1e) σ-bond. Recently, Dronskowski and coworkers have developed an orbital-based method to quantify the chemical bonding mechanisms[54]. A map using the normalized Löwdin charge as the x-axis and the two times integrated crystal orbital bond index (ICOBI) as the y-axis has also been plotted. Both the density-based and orbital-based calculation schemes create very similar maps[55]. Indeed, this is of no surprise because the orbital-based and density-based calculations are applied to the same computed wave functions. Thus, the map can be used as a tool to design materials with desirable properties. As displayed in Fig. 7, we note that apart from the yellow circles highlighted sesqui-chalcogenides ($\beta$-$As_2Te_3$, $Sb_2Te_3$, and $Bi_2Te_3$), there are more candidate tellurides such as GeTe and SnTe (marked in light green circles) in the green shaded MVB region, implying common intrinsic physical properties of these compounds conducive to high thermoelectric performance[29,32]. At the same time, a plethora of non-MVB tellurides, such as ZnTe, CdTe (tagged by light

purple circles), HgTe, and CaTe, are found in the covalent or ionic regions. Thus, we need to select at least two categories of tellurides with different bonding mechanisms to illustrate the predictability of the ES-ET map for advancing the thermoelectric performance of Te. Herein, as a case study, we first used four representative dopants (Ge, Sn; Zn, Cd) to prepare the solely doped Te materials with a nominal dopant concentration of 2%. We fixed the dopant content for ease of demonstration and comparison. According to the combined analysis of XRD patterns, SEM-EDS mappings, and APT results (Figs. S8b, S16–18, S20 (left), Supplementary Table S1, and Fig. 8f–h(top)), all four samples possess dual-phase microstructures with corresponding telluride precipitates embedded in the Te matrix.

As expected, the electrical transport properties of the Te composites with GeTe and SnTe precipitates are superior to those containing ZnTe and CdTe precipitates (see Fig. 8a–c). For example, the difference of electrical conductivity σ at 300 K is about 18 times, from 101.1 S cm$^{-1}$ for $Te_{0.98}Sn_{0.02}$ composite to 5.5 S cm$^{-1}$ for $Te_{0.98}Zn_{0.02}$ composite. Both $Te_{0.98}Ge_{0.02}$ and $Te_{0.98}Sn_{0.02}$ show a metal-like behavior, that is, σ decreases with increasing T in 300–450 K. Moreover, a maximum power factor (PF) up to 10.5 μW cm$^{-1}$ K$^{-2}$ at around 300 K is observed in the $Te_{0.98}Ge_{0.02}$ sample even though the content of Ge could not have been optimized, comparable with the room-temperature values of the As-, Sb- and Bi-containing Te materials. This value is the highest among the reported non-pnictogen-doped Te thermoelectrics[56–58]. Additionally, Ge-/Sn-doped Te composites exhibit slightly lower thermal conductivity than that of Zn-/Cd-doped Te throughout the measured T range (Figs. 8d, e, and S21). MVB tellurides (GeTe, SnTe) usually possess soft chemical bonds and strong lattice anharmonicity, which engenders a low thermal conducting behavior[41]. Here, the effect of MVB precipitates on lowering the lattice thermal conductivity is not as pronounced as on increasing the electrical conductivity. This is because Te already shows an intrinsic low $\kappa_L$ due to its heavy constituent element and quasi-one-dimensional chain structure[59,60], as well as the small fraction of interface phonon scattering. Combining the measured σ, S, and $\kappa_{tot}$, the thermoelectric figure-of-merit zT values for those four Te composites and pure Te are plotted in Fig. 8e. The average $zT_{300-600\ K}$ value of 0.23 for representative $Te_{0.98}Ge_{0.02}$ is larger by about 7-fold and 14-fold of that for $Te_{0.98}Zn_{0.02}$ and pristine Te, respectively. It should be mentioned that zT values could be further improved by fine-tuning the content of dopants. These observations are further supported by the bond-breaking behavior of precipitates in APT measurements, where GeTe and SnTe show high PME (indicative of MVB) while ZnTe and CdTe show low PME (see Fig. 8f–h (bottom) and Fig. S20a (right)). Finally, we evaluated the effect of 22 other dopants across the periodic table on electronic transport. As seen from Supplementary Figs. S22 and S23, analogous to covalent ZnTe and CdTe, these dopant-induced telluride precipitates (Figs. S16, S18, and S19) have little influence on the electronic transport properties. This validates the hypothesis that only MVB telluride precipitates can markedly improve Te thermoelectrics. Hence, this work and, in particular, the map in Fig. 7 provides clear design rules for Te-based thermoelectrics.

Spin theory and topological insulator theory also show considerable promise in identifying high-performance thermoelectrics. The interplay among heat, charge, and spins, which is called spin caloritronics, provides freedom to tune the performance of thermoelectrics[61]. For example, the thermoelectric properties of GeTe can be significantly improved by tuning the strength of Rashba spin splitting[62]. The presence of large spin–orbit coupling (SOC) is also crucial for topological insulators. Many topological insulators such as $Bi_{1-x}Sb_x$, $(Bi,Sb)_2(Te,Se)_3$, and SnTe have long been considered promising thermoelectric materials since they favor the same material features, such as small bandgaps and low phonon frequencies[63–65]. Interestingly, the materials that show Rashba spin splitting effects, such as GeTe, and many topological insulators, such as $Bi_2Te_3$, $Bi_2Se_3$,

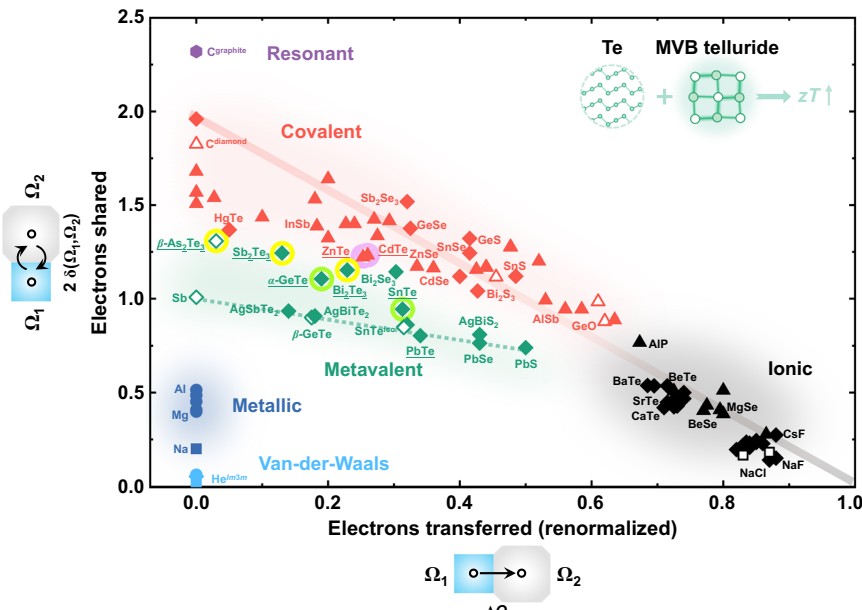

**Fig. 7 | 2D map classifying chemical bonding in solids.** The map is spanned by the renormalized electron transfer (ET) obtained after division through the formal oxidation state (*x*-axis) and the sharing of electrons (ES) between adjacent atoms (*y*-axis). Different colors indicate different material properties and have been related to different chemical bonding mechanisms. Ionic bonding (black-shaded area) is characterized by a significant electron transfer with an ET value larger than 0.6. In covalent solids (red-shaded area), on the contrary, there is only modest ET between atoms, but up to one electron pair (i.e., two electrons) defined by Lewis[84] are shared between neighboring atoms. Metallic bonding (blue-shaded area) is recognized by the delocalization of electrons over several neighbors and, thereby, is characterized via a small or vanishing ET and the sharing of far less than one electron between atoms. MVB materials share about one electron between adjacent neighbors and are characterized via small or moderate ET. Therefore, they are all located in the well-defined (green-shaded) region between covalent and metallic bonding. The red−black solid line characterizes the transition from ideal covalent bonds to perfect ionic bonds. The green dashed line denotes MVB solids with the perfect octahedral arrangement, while all distorted octahedrally coordinated structures are situated above it, characterized by a larger number of ES. Figure replotted and updated upon data exhibited in refs. 29,37. Here, $\beta$-$As_2Te_3$, $Sb_2Te_3$, and $Bi_2Te_3$ are marked with yellow circles; GeTe and SnTe are marked in light green circles; ZnTe and CdTe are tagged with light purple circles. This map can be employed to guide the rational design of high-efficiency Te thermoelectric composites.

and SnTe, prevail in the MVB region of the map[37]. This is evidence for a close link between MVB and topological insulators. Indeed, the characteristic features in the electronic band structures of many topological insulators can be related to the half-filled bands of *p*-orbitals that characterize metavalent solids[66]. Yet, the concept of metavalent bonds also immediately suggests which tuning knobs to play with to tailor performance. A slight charge transfer or Peierls distortions open a small bandgap in metavalent solids[66]. This is beneficial for longitudinal thermoelectrics since a vanishing bandgap would cause strong bipolar effects.

Topological insulator theory focuses on the electronic properties of solids. Yet, MVB theory can also explain phonon transport[40], which is another crucial factor for thermoelectric. Furthermore, MVB theory has demonstrated its ability to predict property trends upon systematically changing ES and ET[29]. Of course, calculating and designing ES and ET for multi-component alloys remains a challenge. In this regard, the combination of the spin, topological insulator, and MVB theories could provide a more comprehensive understanding and screening of outstanding thermoelectrics.

## Discussion

In summary, combining comprehensive characterization techniques, we thoroughly reexamine the microstructure of three typical pnictogens (i.e., As, Sb, and Bi) doped Te thermoelectrics. All samples feature a dual-phase structure consisting of a nearly undoped Te matrix as well as dopant-induced telluride precipitates. By performing mesoscopic electrical transport measurements for doped Te lamellae embedded with and without individual precipitates, we demonstrate that these micrometer-scale precipitates are responsible for improved transport properties. This improvement mechanism is in striking contrast to the usual role of dopants[3,7,17,56,67]. Our detailed APT and first-principles studies further reveal that these precipitates employ metavalent bonds and thus possess a property portfolio beneficial to the advancement of the thermoelectric figure-of-merit for bulk Te. Furthermore, the representative $Bi_2Te_3$/Te heterophase structure is shown to possess a clean, coherent interface with favorable type-I valence band alignment ($E_{VBO}$ of 0.04 eV), giving rise to efficient charge transfer across the interface. These findings present a new point of view for understanding the microscopic origin of the thermoelectric performance of trace-level-doped Te semiconductors. More importantly, we further screen GeTe and SnTe as MVB precipitates under the guidance of the quantum-mechanical-based chemical bonding map and successfully develop novel Te thermoelectric composites, $Te_{0.98}M_{0.02}$ (M = Ge and Sn). Ultimately, the combination of our experimental and conceptual approaches offers a strategy to tailor the relationship between defect chemistry and transport properties of other multiphase systems.

## Methods
### Synthesis
Polycrystalline singly doped Te samples were prepared via melting the stoichiometric compositions of high-purity Te (Sigma Aldrich, 99.999%) and elemental dopants (Sigma Aldrich, 99.999%) such as As, Sb, Bi, Ge, Sn, Zn, and Cd, which were sealed in the graphite-coated quartz tubes with a residual pressure below -$10^{-4}$ Torr. The admixtures were slowly raised to 1323 K in 10 h, dwelled for 20 h, quenched in cold water, then annealed at 673 K for 48 h, and finally, cooled down within the furnace to room temperature. The resulting ingots were hand-grounded into fine powders in an Ar-filled glove box and consolidated by spark plasma sintering (SPS-211Lx) at 663 K for 5 min under axial compressive stress of 45 MPa to obtain highly densified pellets.

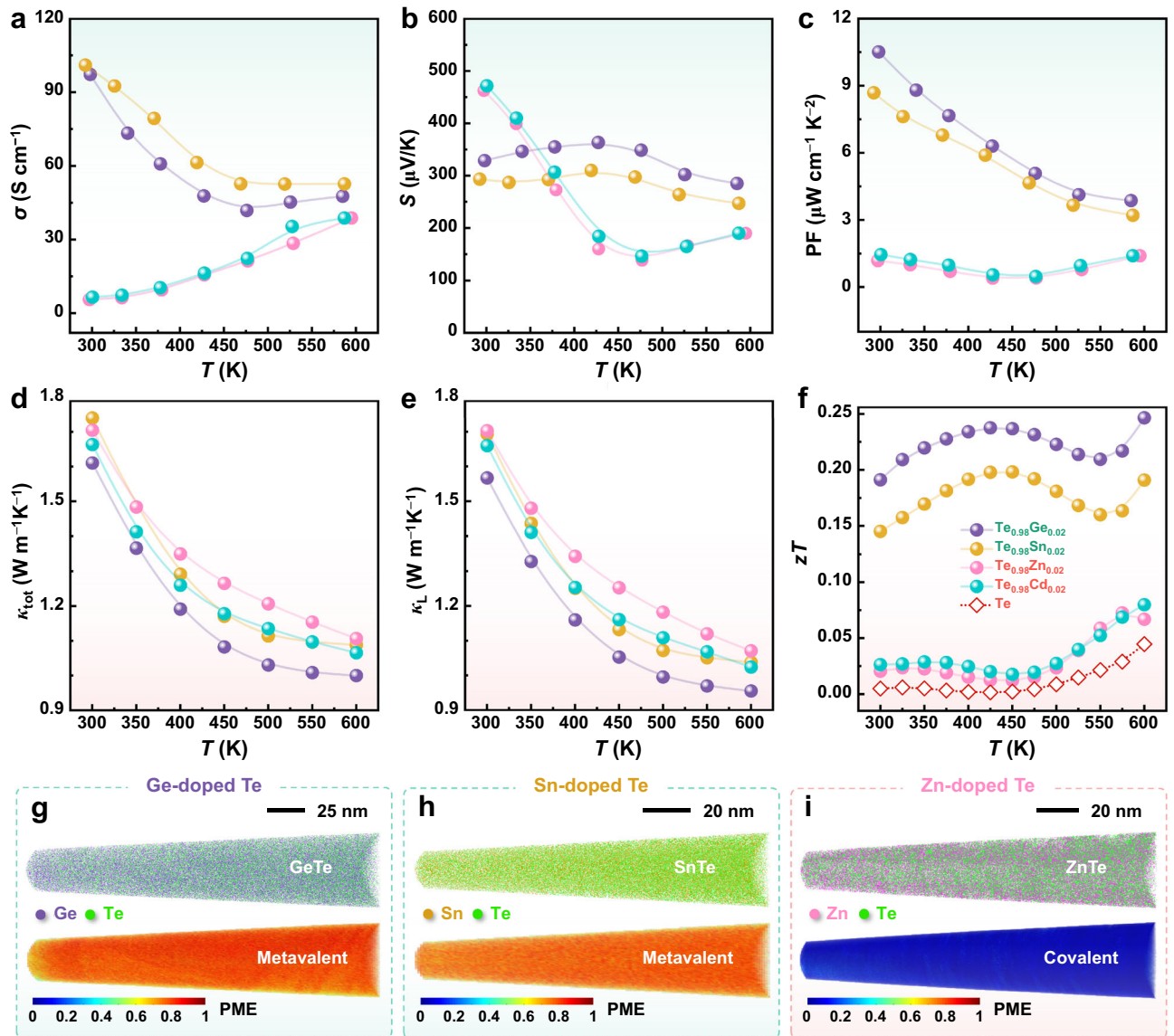

**Fig. 8 | Comparisons on the thermoelectric properties of Te composites with MVB and non-MVB tellurides.** Temperature dependences of **a** electrical conductivity ($\sigma$), **b** Seebeck coefficient ($S$), **c** power factor (PF), **d** total thermal conductivity ($\kappa_{tot}$), **e** lattice thermal conductivity ($\kappa_L$), and **f** thermoelectric figure-of-merit ($zT$) for samples $Te_{0.98}M_{0.02}$ (M = Ge, Sn, Zn, and Cd). The unified doping content of 2% was applied here for ease of verification of our proof-of-principle. **g–i** 3D APT reconstruction and the corresponding PME maps of different telluride precipitates in doped Te materials. The GeTe and SnTe precipitates present high PME values corresponding to metavalent bonding, while the ZnTe precipitate exhibits a low PME value corresponding to covalent bonding. Note that the size of these precipitates is larger than the field of view of APT measurements, and thus the interface is not included.

## Fabrication of mesoscopic transport device

The micro-scale target lamellar crystals were separated from the polycrystalline Te pellets prepared by the above-mentioned melting-quenching-annealing-sintering procedures. The whole process was finished in a dual-beam FIB microscope (Helios NanoLab 650; FEI) using Ga as an ion source, based on the site-specific "lift-out" method[24]. Briefly, three rectangular areas around the region of interest are milled at a voltage of 30 kV using a rectangular scanning pattern at a beam current of 0.79 nA until the trenches are a few micrometers deeper than the intended lamella height. Then an angled cut was made on either side of the sputter-coated Pt layer, and one end was cut free and attached to a micromanipulator (Omniprobe) with the aid of the Pt gas injection system. After that, the FIB-prepared lamella was glued to the Cu grid and further cleaned and reshaped using the cross-cleaning function of the FIB (2 kV, 4 pA). Finally, the desired lamella with the sizes of 2–3 μm in width, 1–2 μm in thickness, and 10–12 μm in length was transferred again via the micromanipulator and then assembled on the SiO₂/Si substrate with Au electrode contacts. Notably, these device electrodes (Ti/Au, 5/25 nm) were defined by electron-beam lithography and e-beam evaporation in an ultra-high vacuum (UHV) chamber (<10⁻⁸ Torr), where the Ti metal is employed as an adhesion layer to enhance the stickiness between SiO₂ and Au. The channel width between the voltage electrodes was 4 μm throughout the fabrications. The sizes of the contacts with the lamella sample for current and voltage electrodes were kept at 200 nm and 1 μm, respectively.

## APT measurement and analysis

The APT tip-shaped specimens were prepared using a standard "lift-out" protocol[68] using a dual beam SEM/FIB, and the final cleaning voltage/current was 2 kV/0.25 nA to remove the surface contamination and gallium implantation. The APT measurement was carried out using a local electrode atom probe (LEAP™ 4000X Si; Cameca) in a laser mode under a high vacuum of ≈10⁻¹¹ Torr. Laser pulses of 200 kHz frequency, 355 nm wavelength, 20 pJ power, and 10–20 ps pulse

duration were applied. The base temperature of the specimen was kept at 40 K; the average evaporation rate was 0.01 ions per pulse; the flight path of ions was 160 mm; and the detection efficiency was 50%. The reconstruction of the 3D atom maps was performed using the visualization and analysis software package IVAS 3.8.0 and AP Suite 6.3 (Cameca). A detailed analysis of "multiple events" was conducted using the in-house developed EPOSA software based on the MatLab platform and the original dataset recorded by the commercial software AP Suite 6.3[69]. It enables precise 3D statistics of the probability of multiple events. The raw dataset includes information on the multiplicity of ions detected within each laser pulse, as well as the X, Y, and Z coordinates of each ion. Thus, a 3D map showing the PME value through the whole sample can be generated by calculating the ratio of multiple events to the total events in e.g. a cube of $0.5 \times 0.5 \times 0.5$ nm$^3$.

### Materials characterizations

Powder X-ray diffraction patterns of Te samples were collected at Rigaku SmartLab SE diffractometer equipped with a Cu K$_\alpha$ source ($\lambda = 1.5418$ Å). The polished surfaces of samples were observed by scanning electron microscopy (JSM-7900F; JEOL) equipped with energy-dispersive X-ray spectroscopy (EDS). (Scanning) transmission electron microscopy (TEM/STEM) observations were carried out in an FEI Themis Z. STEM foils were prepared by a standard procedure including cutting, grinding, dimpling, polishing, and Ar ion-milling (Gatan PIPS Model691). Optical absorption spectra were measured at 300 K using a Bruker Optik GmbH INVENIO R FTIR spectrophotometer ($k = 4000-400$ cm$^{-1}$) equipped with an integrated sphere and a Perkin-Elmer Lambda 850 + UV-vis spectrophotometer ($\lambda = 800-250$ nm), respectively. We also performed Ultraviolet photoelectron spectroscopy (UPS) analyses using PHI 5000 VersaProbe III with a He I$\alpha$ source (21.22 eV) and an applied negative bias of 9 V.

### Thermoelectric properties measurements

For mesoscopic electrical transport measurements, longitudinal resistance was measured in a Hall bar device with a four-terminal configuration based on a physical property measurement system (PPMS DynaCool; Quantum Design). Keithley 2400 source meter was used for DC measurement. A driving current of 1 μA was applied to avoid both the Peltier and the Joule heating effects. Variable temperatures from 2 to 300 K and magnetic fields swapped from −9 to 9 T were achieved under $10^{-4}$ Torr. The temperature equilibrium of the lamella specimen was checked by monitoring the specimen resistivity for stability over time at a fixed temperature. For bulk transport measurements, the electrical conductivity ($\sigma$) and Seebeck coefficient ($S$) were simultaneously measured on a JouleYacht Namicro-3L instrument in a high vacuum atmosphere, using a standard differential voltage/temperature technique and direct-current four-probe method. The thermal diffusivity ($D$) was acquired with Netzsch LFA 467 via the laser flash method. The total thermal conductivity was calculated using $\kappa_{tot} = D\rho C_p$, where the density ($\rho$) was estimated by the Archimedes principle, and the specific heat capacity ($C_p$) was derived from the Dulong-Petit law. The lattice thermal conductivity ($\kappa_L$) is determined by subtracting the electronic thermal conductivity ($\kappa_e$) from $\kappa_{tot}$. Here, $\kappa_e$ can be calculated based on the Wiedemann-Franz law ($\kappa_e = L\sigma T$), where $L$ is the Lorenz number estimated using the equation: $L = 1.5 + \exp(-|S|/116) \times 10^{-8}$ V$^2$ K$^{-2}$ [70]. The measurement uncertainties in $\sigma$, $S$, and $\kappa_{tot}$ were within 5%, 8%, and 5%, respectively. The Hall coefficient ($R_H$) at 300 K was investigated under a reversible magnetic field (1.5 T) using the van der Pauw method (CH-100). The Hall carrier concentration ($n_H$) was determined by $n_H = 1/(eR_H)$.

### Density functional theory calculations

All density functional theory (DFT) calculations were carried out using the Vienna Ab initio Simulation Package (VASP)[71]. The projector-augmented wave (PAW)[72] method was utilized to describe the core-valence electron interaction. The exchange correlation between electrons was treated by the generalized gradient approximation (GGA) formulated by Perdew–Burke–Ernzerhof (PBE)[73]. A dispersion correction of total energy using the DFT-D3 method[74] was adopted to process the van der Waals interactions. The plane-wave cutoff energy was set to 250 eV. To avoid artificial interactions between periodic images, a vacuum layer of 15 Å was added perpendicular to the slab models. In addition, dipole correction[75] was employed to improve the convergence. These atomic structure models were relaxed until the residual forces on the atoms declined to <0.02 eV/Å. The Brillouin zone integration was performed using $12 \times 12 \times 12$, $12 \times 12 \times 1$, $10 \times 2 \times 1$ Gamma-centered k-mesh for Te unit cell, Bi$_2$Te$_3$ unit cell and (102) Te/ (001) Bi$_2$Te$_3$ heterostructure, respectively.

### Reporting summary

Further information on research design is available in the Nature Portfolio Reporting Summary linked to this article.

## Data availability

The authors declare that the data supporting the key findings of this study are available on reasonable request.

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

## Acknowledgements

This work was supported by the National Natural Science Foundation of China (No. 52202277), the Special Project of Science and Technology Cooperation and Exchange of Shanxi Province (No. 202104041101007), and the Fund for Shanxi "1331" Project. Y.Y. acknowledges the financial support under the Excellence Strategy of the Federal Government and the Länder within the ERS RWTH Start-Up grant (Grant No. StUpPD_392-21).

## Author contributions

Y.Y. and D.A. conceived the project. Y.Y., X.-M.Z. and M.W. supervised the project. D.A., W.F., W.W. and S.C. synthesized the samples and performed the thermoelectric transport property measurements. S.Z. and Y.Y. prepared the lamellae specimens while R.W. and H.Z. performed the PPMS measurements. S.Z., Y.Y. and O.C-M. performed APT measurements and data analyses. D.A. and X.Z. carried out (S)TEM measurements and data analyses. W.Y. and D.A. performed the spectroscopy experiments. D.A. and X.-M.Z. contributed to the DFT calculations. D.A. and Y.Y. wrote the draft. M.W. edited the manuscript. All authors commented on and approved the submission of this manuscript.

## Funding

## Competing interests

The authors declare no competing interests.
