## [Peer Review File · Nature Communications]

Metavalently bonded tellurides: the essence of improved thermoelectric performance in elemental TeREVIEWER COMMENTS

Reviewer #1 (Remarks to the Author):

This manuscript, titled "Metavalently bonded tellurides: Unveiling the fundamental aspects of enhanced thermoelectric performance in elemental Te," explores the realm of MVB thermoelectric materials using cutting-edge characterization techniques such as APT and PPMS. The study provides a comprehensive exploration and highlights exceptional characterization methods, enabling a deeper comprehension of the fundamental nature of thermoelectric transport. However, compared to previous reports on MVB and APT, this manuscript does not offer much novelty. Some comments are reported below:

- (1) In this manuscript, the description regarding the essence of MVB materials is limited. It would be beneficial to provide more detailed information on the nature of MVB, specifically whether it involves a coexistence of covalent and metallic bonding, similar to the typical thermoelectric material PbTe. Furthermore, it would be helpful to elaborate on how advanced characterization techniques such as APT demonstrate that the material is valence-bonded. Providing specific details on the experimental results or analysis methods employed in the study would enhance the clarity and credibility of the findings.
- (2) In the context of thermoelectric materials, the thermal performance is just as crucial as the electrical performance. It is worth noting that several materials, such as Ag₉GaSe₆, Ag₈SnSe₆, and AgSbTe₂, have demonstrated promising thermoelectric properties due to their low intrinsic thermal conductivity. While this manuscript primarily emphasizes the characterization of the electrical properties of the material, it is important not to overlook the impact of MVB on thermal conductivity. Incorporating a discussion or analysis of the influence of MVB on thermal conductivity would provide a more comprehensive understanding of the material's thermoelectric performance.
- (3) The manuscript mentions the testing of heterojunctions. However, heterojunctions are typically connected by van der Waals bonds. The main focus of this manuscript is MVB. Why explore heterojunctions and what role do they play in explaining MVB?
- (4) In comparison to the articles published in *Advanced Materials* (Adv. Mater. 2021, 33, 2102356), it is observed that Figure 7 of this manuscript does not present a significant number of potential new thermoelectric materials. Instead, it primarily showcases traditional thermoelectric materials that have already been reported. The MVB theory, as presented in this study, may not demonstrate substantial effectiveness in the discovery of novel thermoelectric materials. However, it is worth noting that alternative theories such as spin theory and topological insulator theory have shown more promise in facilitating the identification of new materials with desirable thermoelectric properties. These theories have demonstrated greater efficacy in guiding the search for innovative thermoelectric materials. Considering this, it would be valuable to discuss the limitations of the MVB theory in discovering novel thermoelectric materials and highlight the potential benefits of exploring other theoretical frameworks such as spin theory and topological insulator theory. By acknowledging the strengths and weaknesses of different approaches, the manuscript can provide a more comprehensive understanding of the current landscape of thermoelectric material research.
- (5) In quantum mechanics, the energy levels of electrons in orbitals are fundamentally determined by the probability of electron presence within those orbitals. Bonding orbitals and antibonding orbitals correspond to the addition and subtraction of wave functions, respectively. In the context of quantum mechanics, bonding involves an increase in electron cloud density between two bonded atoms, facilitating the formation of a chemical bond. Therefore, the explanation of MVB appears to be closely associated with orbital theory. Specifically, we can approach the bonding mechanism of MVB from the perspective of wave

functions. By studying the characteristics and interactions of wave functions, we can gain insight into the influence of MVB on the formation of chemical bonds and material properties. Overall, the revised manuscript will be re-reviewed to determine whether it meets the criteria for publication in the journal.

Reviewer #2 (Remarks to the Author):

Authors have reported the thermoelectric performance of elemental Te, which has been proposed to improve by pnictogen dopants in bulk Te. The results obtained are interesting, however, I have following concerns in the DFT based calculations which need to be addressed:

1. Since pnictogens (As, Sb, Bi) have been used as a dopant in bulk Te, the effect of spin-orbit coupling in the DFT calculations may influence the electronic structure and hence the electrical transport properties reported in the manuscript.
2. In the thermoelectric calculations, it is not clear how author obtain electrical transport properties? How relaxation time was estimated? How various scattering mechanism were handled as these drastically changes the thermoelectric coefficients (electrical conductivity, Power factor etc.) and hence the value of zT ?
3. Lattice thermal conductivity plots may be provided as it gives more accurate theoretical description of thermoelectric performance of materials.
4. Authors may refer following recent papers for the thermoelectric calculations: Phys. Rev. B, 2021, 104, 125115; Phys. Rev. B, 2022, 106, 115105; Nanoscale, 2023,15, 5964-5975.

REVIEWER COMMENTS

Reviewer #1 (Remarks to the Author):

This manuscript, titled "Metavalently bonded tellurides: Unveiling the fundamental aspects of enhanced thermoelectric performance in elemental Te," explores the realm of MVB thermoelectric materials using cutting-edge characterization techniques such as APT and PPMS. The study provides a comprehensive exploration and highlights exceptional characterization methods, enabling a deeper comprehension of the fundamental nature of thermoelectric transport. However, compared to previous reports on MVB and APT, this manuscript does not offer much novelty. Some comments are reported below:

General Response: We are very grateful for your positive evaluation and insightful comments. It is pleasing to read that our *study provides a comprehensive exploration and highlights exceptional characterization methods, enabling a deeper comprehension of the fundamental nature of thermoelectric transport*. We are also glad that you appreciate the *use of cutting-edge characterization techniques such as APT and PPMS*. At the same time, you are concerned that our *manuscript does not offer much novelty, compared to previous reports (we have written) on MVB and APT*.

This last statement shows that we have not managed to get the key message of our manuscript across. Thanks for your open words! We have made a significant number of changes to the manuscript as described in more detail in the following. Here, we only would like to summarize that doping is presumably the oldest strategy to improve thermoelectrics, already established more than 50 years ago. In the present study, doping is accomplished with a surprising twist. There is no single element that we can dope **into the Te host** to achieve attractive thermoelectric properties. Nevertheless, for a well-defined range of doping elements, we form **secondary phases** causing a good thermoelectric performance. This immediately raises two questions: which doping elements lead to these beneficial secondary phases and how do they help to achieve good thermoelectric properties in Te? Both answers are now addressed more clearly in the revised manuscript. In this context, we also relate these findings to metavalent bonding.

(1) In this manuscript, the description regarding the essence of MVB materials is limited. It would be beneficial to provide more detailed information on the nature of MVB, specifically whether it involves a coexistence of covalent and metallic bonding, similar to the typical thermoelectric material PbTe. Furthermore, it would be helpful to elaborate on how advanced characterization techniques such as APT demonstrate that the material is valence-bonded. Providing specific details on the experimental results or analysis methods employed in the study would enhance the clarity and credibility of the findings.

Response: It is good that you urge us to explain the relevance of MVB and its importance for thermoelectrics. Indeed, a new concept such as metavalent bonding can quickly lead to misunderstandings and misinterpretations. Therefore, it is very helpful that you

mention a simple thermoelectric like PbTe as a prominent example. In our map [Raty et al. *Adv. Mater.* **2018**, 31, 1806280], PbTe indeed is situated between metavalent and covalent regions. However, it is not described by a coexistence of covalent and metallic bonds, but more adequately by the **competition between covalent and metallic bonds**. This can also be seen by the unique bond rupture in metavalent solids, as we discuss in detail in a paper submitted yesterday to arXiv [arXiv:2403.04093]. Here, we only show the figure that is discussed and explained in this manuscript, which shows that APT can locally (!) distinguish between metallic, covalent and metavalent bonds. As shown in Figure R1, these bond types can be well separated in a map spanned by two coordinates, the “probability to form molecular ions” (PMI) and the “probability to form multiple events” (PME). Please find a detailed explanation for this difference in our manuscript on arXiv, which shows that the bond rupture in PbTe is not compatible with a mixture of metallic and covalent bonds.

Figure R1. *PMI vs. PME map*. Several classes of materials can be distinguished based on their bond rupture in the atom probe: metals (in blue), metavalent solids (in green), and covalently bonded compounds (in red). Figure taken from Ref [Cojocaru-Mirédin et al., *Atom probe tomography: a local probe for chemical bonds in solids*, arXiv:2403.04093].

Revisions: To clarify this point, we have extended the discussion on the essence of MVB in the revised manuscript on page 10. “Metavalent bonds (MVB) are located in a map spanned by two quantum-chemical bonding descriptors between covalent and metallic bonds [Raty et al. *Adv. Mater.* **2018**, 31, 1806280]. They are best described by a situation where adjacent atoms share half an electron pair (one electron) to form a σ -bond, i.e., a “two center–one electron” (2c–1e) bond [Wuttig et al. *Adv. Mater.* **2023**, 35, 2208485]. This scenario is fundamentally different from ordinary covalent bonds, where adjacent atoms are held together by an electron pair, i.e., a 2c–2e bond. Metavalent bonds also differ from metallic bonds, where the electrons are rather delocalized. Yet, MVB is not a combination of metallic and covalent bonds. It is instead derived from the competition between electron localization as in covalent and ionic bonds and electron delocalization as in metallic bonds [Wuttig et al. *Adv. Mater.* **2018**, 30, 1803777]. This is demonstrated

by the distinctively different properties of materials utilizing these different bonding mechanisms [Wuttig et al. *Adv. Mater.* **2018**, 30, 1803777]. Metavalent solids including PbTe show a unique portfolio of properties, such as a large optical dielectric constant which describes the electronic polarizability, a high Born effective charge which describes the chemical bond polarizability, and a high mode-specific Grüneisen parameter which describes the lattice anharmonicity. This unique property portfolio is not formed by a combination of corresponding properties of covalent and metallic solids.”

We now also include the correlation between PME and MVB on page 10. “In addition, metavalent solids show an abnormal bond-rupture with a large value of PME as illustrated above. In contrast, metals and covalently bonded compounds show PME values below 40%³⁵. The high PME value of metavalent solids is a unique identifier of this bond type. More detailed discussions on the correlation between the high PME and the MVB mechanism can be found in other studies^{27, 31, 35, 36, 38}.”

The analysis of the PME map is now added to the experimental section of the revised manuscript on page 22. “The reconstruction of the 3D atom maps was performed using the visualization and analysis software package IVAS 3.8.0 and AP Suite 6.3 (Cameca). A detailed analysis of “multiple events” was conducted using the in-house developed EPOSA software based on the MatLab platform and the original dataset recorded by the commercial software AP Suite 6.3⁷⁷. It enables precise 3D statistics of the probability of multiple events. The raw dataset includes information on the multiplicity of ions detected within each laser pulse, as well as the X, Y, and Z coordinates of each ion. Thus, a 3D map showing the PME value through the whole sample can be generated by calculating the ratio of multiple events to the total events in e.g. a cube of $0.5 \times 0.5 \times 0.5 \text{ nm}^3$.”

(2) In the context of thermoelectric materials, the thermal performance is just as crucial as the electrical performance. It is worth noting that several materials, such as Ag₉GaSe₆, Ag₈SnSe₆, and AgSbTe₂, have demonstrated promising thermoelectric properties due to their low intrinsic thermal conductivity. While this manuscript primarily emphasizes the characterization of the electrical properties of the material, it is important not to overlook the impact of MVB on thermal conductivity. Incorporating a discussion or analysis of the influence of MVB on thermal conductivity would provide a more comprehensive understanding of the material's thermoelectric performance.

Response: We share your view that for thermoelectric energy conversion the thermal conductivity is as crucial as the electrical performance. Metavalent solids show excellent electrical properties mainly due to their favorable electronic band structures characterized by a large valley degeneracy and low band effective masses. Yet, metavalent solids are also characterized by soft phonons with modest group velocities and a pronounced anharmonicity as now discussed on page 11.

Revisions: “Furthermore, metavalent solids are characterized by half-filled σ -bonds, i.e. a bond order of 0.5 only, compared with a bond order of 1 for ordinary covalent bonds. This leads to weaker and softer bonds, consistent with larger bond lengths. Employing

bond length–bond strength correlations supports the existence of soft bonds in MVB materials. As a result, a low phonon group velocity is obtained in MVB materials³⁸. On the other hand, the delocalized system of *p*-electrons in MVB materials is unstable against transverse optical modes, inducing a softening and large anharmonicity of optical phonons⁴⁸. This finding is closely related to the large Born effective charge. As a result, the phonon relaxation time is greatly reduced due to anharmonic phonon scattering. These two factors contribute to the low intrinsic thermal conductivity of MVB materials^{38,48}.”

(3) *The manuscript mentions the testing of heterojunctions. However, heterojunctions are typically connected by van der Waals bonds. The main focus of this manuscript is MVB. Why explore heterojunctions and what role do they play in explaining MVB?*

Response: In our manuscript, the term *heterojunction* describes the interface between Te matrix and telluride precipitates (such as Bi₂Te₃). These sesqui-chalcogenides have long been considered to employ van der Waals bonds between Te layers. Yet, recent studies reveal that the interactions between these layers are significantly stronger than expected for ordinary vdW bonds [Cheng et al. *Adv. Mater.* **2019**, 31, 1904316; Zhang et al. *Adv. Sci.* **2023**, 10, 2300901]. The significance of this finding and its relevance for the composite materials is now discussed extensively in the manuscript.

Revisions: We have replaced “heterojunctions” with “interfaces” in the revised manuscript to avoid misunderstandings. We now also emphasize the importance of these interfaces on page 12. “The superior thermoelectric properties of MVB tellurides are still insufficient to explain the overall performance improvement since the volume fraction of these precipitates is rather low for a nominal composition of 2% dopants. Therefore, the interface between the matrix and the precipitate must play an important role in the transport properties of the entire sample”. Indeed, Atomic-resolution TEM micrographs in Fig. 5 prove that the interface between Te and Bi₂Te₃ is perfectly matched, revealing a coherent structure. Such a good registry requires strong interactions across the interface, incompatible with weak van der Waals bonds. In typical vdW layered solids, stacking faults can easily be formed at the interface, which we do not observe here [Mio et al. *Adv. Funct. Mater.* **2019**, 29, 1902332]. The strong coupling between the dopant phase and the Te host requires metavalent bonds in the sesqui-chalcogenide precipitate. This is conducive to good charge transport and thus crucial to improving the properties of the composite material.

Only metavalent tellurides form such electrically beneficial interfaces with the Te matrix. This conclusion is proven by studying many telluride precipitates whose bond type is predicted in the map in Fig. 7. SnTe and GeTe form MVB precipitates in Te, significantly improving the zT value. In contrast, covalently bonded precipitates like ZnTe and CdTe only marginally enhance the performance (Fig. 8).

(4) *In comparison to the articles published in Advanced Materials (Adv. Mater. 2021, 33, 2102356), it is observed that Figure 7 of this manuscript does not present a significant*

number of potential new thermoelectric materials. Instead, it primarily showcases traditional thermoelectric materials that have already been reported. The MVB theory, as presented in this study, may not demonstrate substantial effectiveness in the discovery of novel thermoelectric materials. However, it is worth noting that alternative theories such as spin theory and topological insulator theory have shown more promise in facilitating the identification of new materials with desirable thermoelectric properties. These theories have demonstrated greater efficacy in guiding the search for innovative thermoelectric materials. Considering this, it would be valuable to discuss the limitations of the MVB theory in discovering novel thermoelectric materials and highlight the potential benefits of exploring other theoretical frameworks such as spin theory and topological insulator theory. By acknowledging the strengths and weaknesses of different approaches, the manuscript can provide a more comprehensive understanding of the current landscape of thermoelectric material research.

Response: Exploring new MVB materials for applications such as thermoelectrics is a very important research topic. Yet, the present work aims to understand the fundamental role of dopants in tuning the thermoelectric transport properties of elemental Te. We prove that the metavalent telluride precipitate and its well-aligned coherent interface with the Te host are conducive to property improvement. We also illustrate that promising candidate dopants can be predicted by the chemical bonding map. This is very useful for the design of thermoelectrics.

You also suggest searching for new thermoelectric materials and recommend a comparison with material predictions based on spin theory or topological insulator theory. We agree that these theories are also of significance in discovering novel thermoelectrics. Thus, we have extended our discussions on page 19. Specifically, we highlight that many of the topological insulators marked by yellow circles in Figure R2 are located in the MVB region. Details can be found below and in the manuscript.

Figure R2. Correlation between the existence of a topological insulator phase and its mechanism of bonding.

2D map using the basal plane of electrons transferred (ET, i.e., renormalized by the oxidation state) and electrons shared depicting materials which have phases, which are topological insulators by yellow circles. Please see detailed descriptions in Ref [Kooi et al. *Adv. Mater.* **2020**, 32, 1908302].

Revisions: We have revised the manuscript on page 19: “Spin theory and topological insulator theory also show considerable promise in identifying high-performance thermoelectrics. The interplay among heat, charge and spins, which is called spin caloritronics, provides freedom to tune the performance of thermoelectrics [Yang et al. *Nat. Rev. Phys.* **2023**, 5, 466-482]. For example, the thermoelectric properties of GeTe can be significantly improved by tuning the strength of Rashba spin splitting [Hong et al. *Joule* **2020**, 4, 1-14]. The presence of large spin-orbit coupling (SOC) is also crucial for topological insulators. Many topological insulators such as $\text{Bi}_{1-x}\text{Sb}_x$, $(\text{Bi,Sb})_2(\text{Te,Se})_3$, and SnTe have long been considered promising thermoelectric materials since they favor the same material features such as small bandgaps and low phonon frequencies [Müchler et al. *Phys. Status Solidi RRL* **2013**, 7, 91-100; Heremans et al. *Nat. Rev. Mater.* **2017**, 2, 17049; Fu et al. *APL Mater.* **2020**, 8, 040913]. Interestingly, the materials that show Rashba spin splitting effects such as GeTe, and many topological insulators such as Bi_2Te_3 , Bi_2Se_3 and SnTe prevail in the MVB region of the map [Raty et al. *Adv. Mater.* **2018**, 31, 1806280]. This is evidence for a close link between MVB and topological insulators. Indeed, the characteristic features in the electronic band structures of many topological insulators can be related to the half-filled bands of p-orbitals which characterize metavalent solids [Guarneri et al. *Adv. Mater.* **2021**, 33, 2102356]. Yet, the concept of metavalent bonds also immediately suggests which tuning knobs to play with to tailor performance. A slight charge transfer or Peierls distortions open a small bandgap in metavalent solids [Guarneri et al. *Adv. Mater.* **2021**, 33, 2102356]. This is beneficial for longitudinal thermoelectrics since a vanishing bandgap would cause strong bipolar effects.

Topological insulator theory focuses on the electronic properties of solids. Yet, MVB theory can also explain phonon transport [Yao et al. *The Innovation* **2023**, 4, 100522], which is another crucial factor for thermoelectric. Furthermore, MVB theory has demonstrated its ability to predict property trends upon systematically changing ES and ET [Yu et al. *Adv. Mater.* **2023**, 35, 2300893]. Of course, calculating and designing ES and ET for multi-component alloys remains a challenge. In this regard, the combination of the spin, topological insulator, and MVB theories could provide a more comprehensive understanding and screening of outstanding thermoelectrics.”.

(5) *In quantum mechanics, the energy levels of electrons in orbitals are fundamentally determined by the probability of electron presence within those orbitals. Bonding orbitals and antibonding orbitals correspond to the addition and subtraction of wave functions, respectively. In the context of quantum mechanics, bonding involves an increase in electron cloud density between two bonded atoms, facilitating the formation of a chemical bond. Therefore, the explanation of MVB appears to be closely associated with orbital theory. Specifically, we can approach the bonding mechanism of MVB from the perspective of wave functions. By studying the characteristics and interactions of wave*

functions, we can gain insight into the influence of MVB on the formation of chemical bonds and material properties.

Response: In a recent paper published by one of us [Adv. Sci. 2024, 11, 2308578], metavalent bonds have also been discussed from a wave function perspective. This has been possible by utilizing an orbital-based method to quantify chemical bonds [Müller et al. J. Phys. Chem. C 2021, 125, 7959]. Figure R3 (right) illustrates the orbital-based analyses of chemical bonding using the normalized Löwdin charge as the x-axis and the two times ICOBI (integrated crystal orbital bond index) as the y-axis. The results show that both the density-based (Figure R3_left) and orbital-based (Figure R3_right) calculations produce very similar maps. Specifically, different bond types such as metallic, covalent and ionic bonds are located in similar regions in the two maps. In both maps, metavalent solids are characterized by small charge transfers and sharing of about half an electron pair (bond order is $\frac{1}{2}$). This good agreement is no surprise since the orbital-based and the density-based calculations analyze the same computed wave functions.

Figure R3. 2D maps classifying chemical bonding in solids. The map on the left is obtained from a density-based approach, while the map on the right is determined from an orbital-based approach. Please find more details in Ref [Wuttig et al. Adv. Sci. 2024, 11, 2308578].

Revisions: Please see our changes on page 16: “The formation of different bonds between atoms can be attributed to the different behavior of valence electrons such as electron transfer as in ionic bonds or electron pair formation as in covalent and metallic bonds⁴⁴. Using quantum-mechanical calculation tools, the number of electrons transferred (ET) from one atom to its neighboring atom and the value of electrons shared (ES) between two neighboring atoms can be obtained⁶⁶. These two values (ES and ET) are excellent descriptors of chemical bonds. Thus, a 2D map spanned by ES and ET can be established, where different types of chemical bonds are located in well-defined regions of the ES-ET map. MVB prevails in a region with a small ET and an ES value close to one. This is also in line with the nature of MVB as a “two center–one electron” (2c–1e) σ -bond. Recently, Dronskowski and coworkers have developed an orbital-based method to quantify the chemical bonding mechanisms⁶⁷. A map using the normalized Löwdin charge as the x-axis and the two times ICOBI (integrated crystal orbital bond index) as the y-axis has also been plotted. Both the density-based and orbital-based calculation schemes create very similar maps⁶⁸. Indeed, this is of no surprise because the orbital-based and density-based

calculations are applied to the same computed wave functions. Thus, the map can be used as a tool to design materials with desirable properties.”

Overall, the revised manuscript will be re-reviewed to determine whether it meets the criteria for publication in the journal.

Response: Thank you again for your constructive comments and suggestions. We have significantly revised the manuscript and hope that it now meets your criteria for publication in Nature Communications.

Reviewer #2 (Remarks to the Author):

Authors have reported the thermoelectric performance of elemental Te, which has been proposed to improve by pnictogen dopants in bulk Te. The results obtained are interesting, however, I have following concerns in the DFT based calculations which need to be addressed:

General Response: We are grateful for your suggestions to improve the quality of our manuscript. It is pleasing to read that our work is considered interesting. Below, we provide a point-by-point response to your comments.

1. Since pnictogens (As, Sb, Bi) have been used as a dopant in bulk Te, the effect of spin-orbit coupling in the DFT calculations may influence the electronic structure and hence the electrical transport properties reported in the manuscript.

Response: We understand the reviewer’s concern about the underlying effects of spin-orbit coupling (SOC) on the electronic transport properties of doped Te. However, all the electrical transport properties reported in this work were measured experimentally. Thus, the high relevance of SOC for DFT calculations is less crucial in this work.

Generally, for compounds containing heavy atoms, SOC influences the electrical properties such as the bandgap and the spin-splitting effect [Vi et al. *J. Phys. D: Appl. Phys.* **2022**, 55, 505302; Hasani et al. *J. Phys. Chem. Solids* **2023**, 174, 111131]. However, it has been shown by Han et al. that the bandgap of Te will be significantly underestimated if the SOC effect is included in the calculation [Han et al. *J. Phys. Chem. Solids* **2019**, 135, 109114]. Thus, the electronic band structure of elemental Te as shown in the supplementary Fig. S1 does not include the SOC effect. Similar treatments and results have also been reported elsewhere [Peng et al. *Phys. Rev. B* **2014**, 89, 195206].

Nevertheless, we sincerely appreciate the reviewer’s comment about including SOC effects when studying the electrical properties of solids with heavy elements by DFT calculations.

Revisions: We have explained why we did not include the SOC effect in calculating the electronic band structures of Te in the supplementary information Figure S1. “Generally, for compounds containing heavy atoms, SOC influences the electrical properties such as the bandgap and the spin splitting effect [Vi et al. *J. Phys. D: Appl. Phys.* **2022**, 55, 505302; Hasani et al. *J. Phys. Chem. Solids* **2023**, 174, 111131]. However, it has been shown by Han et al. that the bandgap of Te will be significantly underestimated if the SOC effect is included in the calculation [Han et al. *J. Phys. Chem. Solids* **2019**, 135, 109114]. Thus, the electronic band structure of elemental Te as shown in Fig. S1 does not include the SOC effect. Similar treatments and results have also been reported elsewhere [Peng et al. *Phys. Rev. B* **2014**, 89, 195206].”

2. In the thermoelectric calculations, it is not clear how author obtain electrical transport properties? How relaxation time was estimated? How various scattering mechanism were handled as these drastically changes the thermoelectric coefficients (electrical conductivity, Power factor etc.) and hence the value of zT ?

Response: In this work, the values of zT ($zT = S^2 \sigma T / \kappa$) were experimentally determined by measuring the temperature-dependent electrical conductivity (σ), Seebeck coefficient (S), and thermal conductivity (κ). All these measurements were carried out using commercial setups as schematically sketched in Figure R4 and described in detail in the experimental section. In this manuscript, we did not carry out theoretical calculations (based on DFT and the Boltzmann transport theory) of thermoelectric transport properties.

Figure R4. Schematic display of thermoelectric measurements. **a** The longitudinal resistivity ($\rho = 1/\sigma$) of the sample measured by a four-probe method. **b** The Seebeck coefficient determined by measuring the temperature difference across the sample and its resulting voltage difference between the two thermal couples. **c** The Hall coefficient was determined by measuring the transverse Hall voltage generated by

applying a magnetic field (B) perpendicular to the current direction. **d** The thermal diffusivity was measured by the laser flash method. An IR sensor is used to measure the temperature increase as a function of time. Finally, the thermal diffusivity of the sample can be calculated using a correction model by Cowan [Wei et al. *Joule* **2018**, 2, 1-6].

Revisions: We have added more details for the measurement of thermoelectric transport properties in the experimental section of the manuscript on page 23.

3. Lattice thermal conductivity plots may be provided as it gives more accurate theoretical description of thermoelectric performance of materials.

Response: We appreciate the reviewer's suggestions. We have added temperature-dependent lattice thermal conductivity data (κ_L) in Fig. 8e, S3b, and S4c, and discuss these data in the revised manuscript on page 18.

Revisions: We have added the lattice thermal conductivity plot in Fig. 8e of the revised manuscript. Please find corresponding descriptions on page 18 as “Additionally, Ge-/Sn-doped Te composites exhibit slightly lower thermal conductivity than that of Zn-/Cd-doped Te throughout the measured T range (Fig. 8d, 8e, and Fig. S21). MVB tellurides (GeTe, SnTe) usually possess soft chemical bonds and strong lattice anharmonicity, which engenders a low thermal conducting behavior [Lee et al. *Nature Communications* **2014**, 5, 3525]. Here, the effect of MVB precipitates on lowering the lattice thermal conductivity is not as pronounced as on increasing the electrical conductivity. This is because Te already shows an intrinsic low κ_L due to its heavy constituent element and quasi-one-dimensional chain structure [An et al. *ACS Appl. Mater. Interfaces* **2019**, 11, 27788–27797; Peng et al. *Appl. Phys. Lett.* **2015**, 107, 251904], as well as the small fraction of interface phonon scattering.”

4. Authors may refer following recent papers for the thermoelectric calculations: *Phys. Rev. B*, 2021, 104, 125115; *Phys. Rev. B*, 2022, 106, 115105; *Nanoscale*, 2023,15, 5964-5975.

Response: We appreciate your recommendation to extend the list of DFT papers related to thermoelectric transport that are referenced in our manuscript.

Revisions: The suggested references have been added and are highlighted in the bibliography.

REVIEWERS' COMMENTS

Reviewer #1 (Remarks to the Author):

Accept

Reviewer #2 (Remarks to the Author):

Authors have satisfactorily addressed the comments of my previous report, hence, the manuscript can be published in the current form now.

REVIEWER COMMENTS

Reviewer #1 (Remarks to the Author):

Accept.

Response: Thanks.

Reviewer #2 (Remarks to the Author):

Authors have satisfactorily addressed the comments of my previous report, hence, the manuscript can be published in the current form now.

Response: Thank you.